# Using dried blood spots to estimate *Toxoplasma gondii* seroprevalence in pregnant women in Catalonia, Spain, and to serologically diagnose congenital toxoplasmosis

Borja Guarch-Ibáñez [1,2,3☯]*, Ana Argudo-Ramírez[4☯], Clara Carreras-Abad[3,5],
Marie Antoinette Frick[3,6], Daniel Blázquez-Gamero[3,7,8,9,10],
Fernando Baquero-Artigao[3,10,11,12], Judit García-Villoira[4],
the Spanish REIV-TOXO group[¶], Pere Soler-Palacín[6‡], Isabel Fuentes[13‡]*

1 ICS-IAS Pediatric Infectious Diseases Unit, Paediatrics Department, Hospital Universitari de Girona Dr. Josep Trueta, Girona, Spain, 2 Universitat de Girona (UDG), Girona, Spain, 3 Congenital Infections Working Group, Sociedad Española de Infectología Pediátrica (SEIP), Madrid, Spain, 4 Inborn Errors of Metabolism Division-IBC, Biochemistry and Molecular Genetics Department, Hospital Clínic, Barcelona, Spain, 5 Pediatric Infectious Diseases Unit, Paediatrics Department, Hospital Universitario Germans Trias y Pujol, Badalona, Spain, 6 Pediatric Infectious Diseases and Immunodeficiencies Unit. Children's Hospital. Vall d'Hebron Barcelona Hospital Campus, Barcelona, Spain, 7 Pediatric Infectious Diseases Unit. Hospital Universitario 12 de Octubre, Madrid, Spain, 8 Instituto de Investigación Hospital 12 de Octubre (i+12), Madrid, Spain, 9 Universidad Complutense de Madrid, Madrid, Spain, 10 Centro de Investigación Biomédica en Red de Enfermedades Infecciosas (CIBERINFEC), Instituto de Salud Carlos III, Madrid, Spain, 11 Pediatric Infectious Diseases Unit. Hospital Universitario La Paz, Madrid, Spain, 12 Universidad Autónoma de Madrid, Madrid, Spain, 13 Toxoplasmosis and Intestinal Protozoa Unit, Centro Nacional de Microbiología, Instituto de Salud Carlos III, Madrid, Spain

☯ These two authors have contributed equally to this work.
‡ These two authors jointly supervised this work.
¶ A complete list of the members of the REIV-TOXO Study Group is provided in Supporting Information (S1 Appendix).
* bguarch.girona.ics@gencat.cat (BGI); ifuentes@isciii.es (IF)

## Abstract

Recent data on toxoplasmosis seroprevalence in Spain is limited, and pregnancy-screening programs are being discontinued. This study aimed to describe seroprevalence in pregnant women in Catalonia using newborn dried blood samples (DBS) from the Catalonian Newborn Screening (NBS) Program; as well as to assess detection of *Toxoplasma gondii* antibodies in DBS (Anti-*Toxoplasma gondii* ELISA (IgG/ IgM), Euroimmun Lubeck, Germany) as a NBS tool and for retrospective diagnosis of congenital toxoplasmosis (CT) in cases from the Spanish Research Network of Congenital Toxoplasmosis (REIV-TOXO). A total of 3,231 DBS samples from NBS program (September 2022–August 2023) were randomly selected and analyzed, alongside with 30 DBS from CT confirmed cases. The overall seroprevalence of *T. gondii* IgG in pregnant women was 15.5% (95%CI: 14.33-16.85), higher in foreign women 22.6%, particularly Latin American (30.3%). Analysis of DBS from CT cases showed concordance in IgG detection but only 3.3% (1/30) were IgM positive despite 37% (10/27) having positive serum IgM at birth. In conclusion, IgG analysis in

**Data availability statement:** There are ethical restrictions on sharing data set because it contains potentially identifying and sensitive patient information and this restriction is imposed by the Research Ethics Committee. To request data please contact the Research Ethics Committee of Hospital Universitari de Girona Dr. Josep Trueta (ceic.girona.ics@gencat.cat) and Ethics Committee of Instituto de Salud Carlos III (registro.general@isciii.es). There is no third party organization that has access to the complete data included in the article.

**Funding:** This work was supported by the national project FIS AESI PI21CIII/00031 from the Instituto de Salud Carlos III, Ministry of Science and Innovation (ICIII to BGI, AAR, and IFC), and by a private donation from the Bescos Manau family (BMF to BGI and PSP) to support research in congenital toxoplasmosis. DBG was supported by the Ministry of Science and Innovation, Instituto de Salud Carlos III, and FEDER funds through the program "Contratos para la intensificación de la actividad investigadora en el Sistema Nacional de Salud, 2023" (INT23/00039). The funders had no role in study design, data collection and analysis, decision to publish, or preparation of the manuscript.

**Competing interests:** The authors have declared that no competing interests exist.

newborn DBS provides valuable information on pregnant seroprevalence. However, its value for identifying CT cases in retrospective diagnosis or as neonatal screening is poor given the low IgM detection. Thus, prenatal screening remains the most effective approach to identify children at risk.

## Author summary

Toxoplasmosis is a parasitic infection that can cause serious complications during pregnancy if transmitted to the fetus. Although its epidemiological situation in Spain is not well established, prenatal screening programs are being discontinued in some regions. We analyzed more than 3,000 dried blood spot samples collected by the Catalonia Newborn Screening Program to estimate how many pregnant women had been exposed to this infection. We found that 15.5% of women showed evidence of past infection, with higher rates among women born abroad—especially from South America and Africa—confirming a decline versus earlier decades, yet most women remain at risk during pregnancy. We also assessed whether dried blood spot samples could be used to diagnose congenital toxoplasmosis. Specific IgG antibodies were consistently detected, but specific IgM antibodies—used as an early marker in newborns—were rarely found. Thus, dried blood spot samples are unsuitable for neonatal screening or reliable retrospective diagnosis, especially when samples are tested long after storage. Neonatal screening with dried blood spot samples cannot be relied upon to detect all cases of congenital toxoplasmosis. Effective diagnosis should depend on prenatal screening, the only strategy to identify all children at risk and enable timely treatment and follow-up.

## Introduction

Toxoplasmosis, caused by the protozoan parasite *Toxoplasma gondii*, is one of the most prevalent zoonotic diseases that can affect pregnant women (PW) [1–2]. Congenital toxoplasmosis (CT) occurs when a woman acquires primary infection during pregnancy, and the parasite is transmitted across the placenta to the fetus. The infected fetus can present a wide clinical spectrum, ranging from asymptomatic to severe symptomatic forms potentially leading to miscarriage, fetal loss or severe ocular, neurological, and/or systemic involvement at birth. These individuals are also at risk of developing clinical complications and sequelae of toxoplasmosis throughout their lifetime [3–10].

The impact of CT on the population is influenced by a complex interplay of biological, socio-economic, and environmental factors (Fig 1).

The seroprevalence in the community is influenced by several variables, including climatic conditions that support the parasite's survival, hygienic and dietary habits related to food consumption and water quality control, and the control of animal

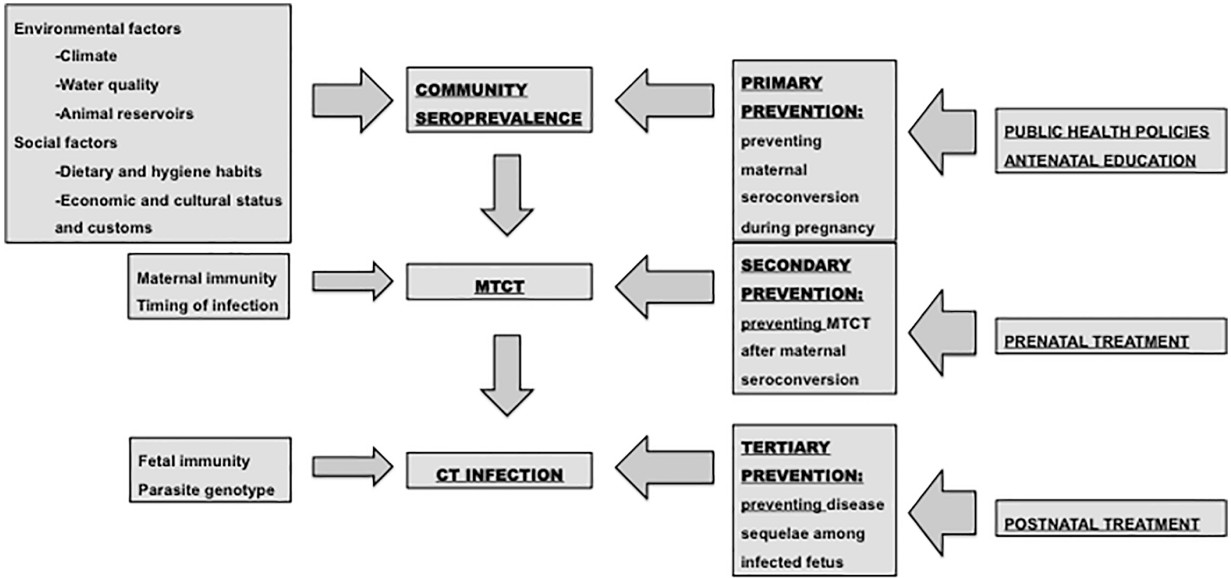

**Fig 1. Current and potential factors involved in congenital toxoplasmosis and preventive methods.** *MTCT = mother-to-child transmission; CT = congenital toxoplasmosis.*

reservoirs [1,6,11,12]. To mitigate the spread of the disease, a variety of public health strategies targeting both animals and humans have been proposed, including antenatal education of PW (Fig 1). While the latter is considered a valuable intervention, its effectiveness remains insufficiently evaluated [13–17]. In Spain, epidemiological studies conducted between 2006 and 2010 showed a seroprevalence in PW between 12% and 23.4%, with a higher prevalence in migrant women (33.8-41.4%) [13,18,19]. A subsequent systematic review and meta-analysis estimated a pooled toxoplasmosis seroprevalence of 24.4% among PW in Spain and a decrease has been observed in the last years [20]. In contrast, recent findings have shown that *T. gondii* is prevalent in wildlife across Spain, with infection rates as high as 85% in wild cats [21–22], which could suggest significant environmental contamination and increased risk for other animals and humans.

Acurately determination of seroprevalence in a specific geographic area is essential for assessing the risk to PW and guiding targeted prevention and control strategies. Traditionally, this has been achieved through the detection of *Toxoplasma* gondii-specific, immunoglobulin G (IgG) antibodies, which appear 2–4 weeks after primary infection and persist lifelong, serving as a reliable marker of past exposure. In contrast, immunoglobulin M (IgM) antibodies appear earlier, often within one week of infection, and could be more indicative of recent or active infection. Importantly, maternal IgG antibodies cross the placenta and are transferred to the fetus, meaning that their detection in neonatal blood samples reflects the maternal serostatus at the end of pregnancy [3–4]. This immunological feature provides an opportunity to leverage newborn dried blood samples (DBS) collected through neonatal screening programs as a surrogate for maternal seroprevalence, enabling indirect but accurate population-level estimates in regions where routine prenatal screening is not implemented [4,23].

The next key factor influencing the incidence and severity of CT is mother-to-child transmission (MTCT). Several well-established factors contribute to MTCT, including maternal immunity, timing of infection and prenatal treatment (Fig 1) [4,24]. Modern studies, such as the one conducted by the Spanish Research Network on Congenital Toxoplasmosis (REIV-TOXO) [25], have emphasized the critical role of prenatal treatment in reducing MTCT, as well as minimizing the risk of clinical complications and long-term sequelae [25–35].

Finally, parasite genotype, fetal immune response, and postnatal treatment each have a critical impact on the outcomes of CT [4]. Current evidence suggests that treatment after birth improves prognosis by reducing the severity of both acute symptoms and sequelae [4,36–40] (Fig 1).

Since toxoplasmosis often presents as an asymptomatic infection during pregnancy, most maternal infections are identified through prenatal serological screening. This approach is also the only way to detect all infants with CT, as many infections are asymptomatic at birth and may not show any abnormalities on prenatal ultrasound [3,4,41]. Recent data from REIV-TOXO registry revealed that prenatal screening successfully identified 92.8% of CT cases followed in the analyzed cohort, either at birth or within the first year of follow-up [25].

Neonatal screening based on the detection of *T. gondii* specific IgM and immunoglobulin A (IgA) in cord blood or on stored DBS samples has been proposed as an alternative to prenatal serological screening [4,42–53]. According to some studies, newborn screening on DBS samples could identify around 75% of cases of CT [6,11,46]. Nevertheless, this approach has been rejected in Europe, since it means that infected women do not receive treatment during pregnancy to prevent MTCT [46]. Moreover, the detection of parasite DNA by polymerase chain reaction (PCR) in DBS samples may contribute to improving the diagnosis of CT, but the possibility of detecting parasite DNA is very low given the small amount of blood and the short period of parasitemia of the *T.gondii* infection. Therefore, its routine implementation has not been widely extended.

We hypothesized that DBS-based IgG detection accurately reflects maternal seroprevalence and that DBS-based IgM detection is insufficient for diagnosing all cases of CT. Therefore, the main objective of the present study was to evaluate the utility of DBS samples obtained through the Catalonia Newborn Screening Program (NBS), for estimating the current seroprevalence of toxoplasmosis in PW in Catalonia –a Northern-East Spanish region where systematic prenatal screening for CT is not currently implemented [54] – through the detection of *T. gondii*-specific IgG antibodies. The secondary objective was to determine the diagnostic value of serological testing (IgM, IgG) of DBS samples collected at birth, either for retrospective diagnosis of the disease or as part of population-based neonatal screening.

## Materials and methods

### Ethics statement

The study received approval from the Health Department of Catalonia (*Departament de Salut, Generalitat de Catalunya*), permitting the investigators to utilize the DBS samples and data obtained from the NBS Program. Additionally, the study was approved by the Hospital Trueta Ethics Committee as the coordinating center (CEIm code 2018.027), by the local ethics committee of each participating center, and by the Carlos III Health Research Institute Ethics Committee (Nº: CEI PI 42_2022). The study was conducted in adherence to the principles outlined in the Code of Ethics of The World Medical Association (Declaration of Helsinki). Written informed consent was obtained from the parents of all participating children included in the REIV-TOXO cohort.

### Population and data collection

DBS samples received as part of the universal Catalonia NBS program between September 2022 and August 2023 were randomly selected (3,231 from 56,168 (5.7%) samples received during the study period) and retrospectively analyzed - one year later at most - in its NBS Laboratory.

Samples with the following characteristics were excluded: collection time before 24 hours, blood product transfusion, and poor quality or blood amount. In the case of multiple gestations, only one newborn was studied.

The minimum sample size was calculated assuming a 24.4% *T. gondii* IgG seroprevalence in Catalonia [21], resulting in 3,150 samples to achieve a precision of ±1.5% (95% two-sided normal asymptotic confidence interval).

Demographics (birth date, date of sample collection, birth weight, gestational age, newborn gender, parents' origin, mother's age and place of residence) were electronically collected from the NBS database. Mothers were classified as

foreign-born based on country of birth, regardless of current nationality. Places of residence were classified as rural or urban areas: a living environment categorized as a rural area was defined as a municipality with a population <5,000 inhabitants or a population density of <150 inhabitants/km$^2$, in accordance with the OCDE classification of rural communities; all others were classified as living in urban areas [55]. The number of inhabitants and population density of the different areas were obtained from the register of municipal census available at *Institut d'Estadística de Catalunya* (last updated December 2023).

The REIV-TOXO registry provided data about CT patients in order to assess the value of *T. gondii*-specific IgG and IgM antibodies at DBS specimens, obtained at birth, as a retrospective diagnostic tool for CT and its usefulness as a NBS for the disease. REIV-TOXO is a national database, which includes 122 hospitals in Spain, and compiles different variables on children affected with CT born in Spain between 1st January 2015 and 1st August 2024, using the Research Electronic Data Capture platform (REDCap; 7.6.5, Vanderbilt University, Nashville, TN, USA). CT was defined in REIV-TOXO as having one or more of the following criteria: positive PCR in amniotic fluid (AF); presence of *T. gondii*-specific IgM or IgA; positive PCR for *T. gondii* in the newborn's blood, urine, cerebrospinal fluid (CSF) or in the placenta; increase of specific IgG *Toxoplasma* levels during the infant's follow-up; or persistence of specific IgG beyond 12 months of life. The DBS samples were obtained after contacting the REIV-TOXO collaborating researchers, who requested the shipment of the preserved DBS sample from the corresponding regional NBS laboratory to the Catalan NBS Laboratory at *Hospital Clínic*, Barcelona, to be processed. Prior written informed consent was obtained from the families. Maternal history, including prenatal treatment received, clinical manifestations, and the presence of *T. gondii*-specific IgM and IgG antibodies in serum at birth, were collected for all CT cases. Serological techniques used to detect *T. gondii*-specific IgM and IgG antibodies varied according to the internal protocol at each facility. Prenatal treatment with spiramycin (SPI) and/or pyrimethamine, sulfadiazine, and folinic acid (PSA) was defined as receiving one or both treatments during pregnancy for at least 28 days.

### Sample testing

DBS samples (4.7-mm diameter spot) were analyzed for *Toxoplasma*-specific IgM and IgG antibodies by enzyme-linked immunosorbent assay (ELISA) (Euroimmun Lubeck, Germany). Samples from the Catalonia NBS Program were specifically tested for *T. gondii*-specific IgG, whereas those from REIV-TOXO patients were analyzed for both *T. gondii*-specific IgG and IgM antibodies.

Briefly, DBS spot samples were eluted into uncoated microplates adding 250μL sample buffer of the ELISA *Toxoplasma gondii* (IgG/IgM) kit for 1 hour at 37ºC, according to the manufacturer's instructions. After mix each eluate, 100μL was transferred into the ELISA plate to perform the analysis. Both classes of *T. gondii* specific antibodies were detected by applying a manual methodology, according to the manufacturer's instructions. Euroimmun Anti-*Toxoplasma gondii* ELISA (IgG/IgM) kits include reagents, calibrator and negative and positive quality controls.

DBS samples were retrospectively analyzed one year later at most for seroprevalence study, and in August 2023 for positive CT confirmed cases from the REIV-TOXO registry, with an interval between sample collection and analysis ranging from 1 to 103 months. Samples for seroprevalence study were stored at a controlled temperature (20±5 ºC) and samples for positive CT confirmed cases were stored under variable conditions, which may have affected their real-time diagnostic sensitivity. Five samples from the Newborn Screening Quality Assurance Program anti-*Toxoplasma* IgM Antibodies in Dried Blood Spots Proficiency Testing Program (TOXOPT) provided by the CDC (Centers for Disease Control and Prevention, Atlanta, USA) were included in the study as controls of the technique (S1 Table). External quality control samples for IgG-*Toxoplasma* in DBS were not available. Additionally, ten serum samples with known results for both IgG and IgM-*Toxoplasma* were analyzed in order to perform the kit sensitivity (S1 and S2 Tables). All external positive controls were validated through the serological method employed in the present study.

Results were evaluated semiquantitatively by calculating the ratio of the extinction value of the control or patient sample over the extinction value of the appropriate calibrator. The observed ratios of IgG and IgM antibodies were interpreted

according to the manufacturer's instructions, as follows: < 0.8 negative; ≥ 0.8 to <1.1 borderline; ≥ 1.1 positive. Because a second sample was not available for confirmation, borderline *Toxoplasma*-specific antibody results were reclassified as negative for all stadistical analyses. This conservative approach was adopted to avoid overestimating seroprevalence.

## Statistical analysis

Qualitative variables were expressed as absolute and relative frequencies, whereas quantitative variables are expressed as the mean and standard deviation or the median and interquartile range (IQR), depending on the distribution of the variable. The comparison between variables was carried out using the chi-square test or the Wilcoxon test, depending on the variables studied.

Finally, a univariate and multivariate logistic regression model were performed in order to determine whether the different characteristics analyzed have an effect on the probability of obtaining a positive result. The significance level was set at 95% (p < 0.05) for all analyses, and all analyses were performed using the software SAS v9.4 (SAS Institute Inc., Cary, NC, USA). Sensitivity and specificity of the *T. gondii*-specific IgM assay in DBS samples were calculated and their 95% confidence intervals.

## Results

### *T. gondii* seroprevalence in pregnant women in Catalonia

A total of 3,200 DBS samples were analyzed (31 individuals were excluded from twin pregnancies) (Fig 2A), of whom 99,8% (3,152) came from newborns whose mothers were residents of 397 municipalities in Catalonia. The median age of PW was 33 years (IQR 29–37). The most frequent age group was between 25 and 35 years (50.5%). Of the DBS samples analyzed, approximately two-thirds corresponded to children of mothers born in Spain. These demographic values were similar to those observed in the general population, which were calculated based on the corresponding number of live births in Catalonia during the study period according to official records from our NBS Program (n = 98,857; S3 Table).

After Europe, the most frequent continents of maternal origin were South America and Africa. Most of PW came from urban areas of residence. The main demographic characteristics are shown in Table 1. Slight variations in the denominators across demographic variables reflect missing data in the NBS database.

The overall mean of *T. gondii*-specific IgG in DBS prevalence was 15.5% (95% CI: 14.3-16.8) (Table 1). Amongst the Spanish PW the seroprevalence was 11.8% (95% CI: 10.4-13.2), and amongst foreigners was 22.6% (95% CI: 20.2-25.1) with a statistically significant difference (p < 0.01). Amongst foreigners, the highest seroprevalence was observed in South American countries (30.3%) with a statistically significant difference for foreign PW (p < 0.01). South American PW showed a higher probability of presenting positive *T. gondii*-specific IgG compared to Europeans (OR: 3.0, $IC_{95\%}$=[2.4 - 3.8].

The median age of the PW was statistically higher in the group with positive *T. gondii*-specific IgG (34 versus 33 years; p < 0.01). The logistic regression model showed a lower risk of having a positive IgG for *T. gondii* in DBS among PW aged 25–35 years compared to those older than 35 years (OR = 0.6; $IC_{95\%}$ [0.5-0.8]).

No statistically significant differences were observed between the area of residence (rural or urban) (p = 0.95).

The same significant results for *T. gondii*-specific IgG positivity were also obtained from the multivariate analysis adjusted for the continent of origin and age (p < 0.001).

### Neonatal CT screening based on the detection of *T. gondii*-specific IgM and IgG antibodies in DBS samples

From 63 confirmed CT patients included in REIV-TOXO registry, 30 were eventually included in this study (Fig 2B). Catalonia (19 cases) was the autonomous community that provided the highest number of cases, followed by the Community of Madrid, Valencian Community and Galicia, with 7, 3 and 1 case respectively. In case of no DBS sample was obtained, different reasons were reported: the lack of preservation at the corresponding regional NBS laboratory (12/33; 36.3%),

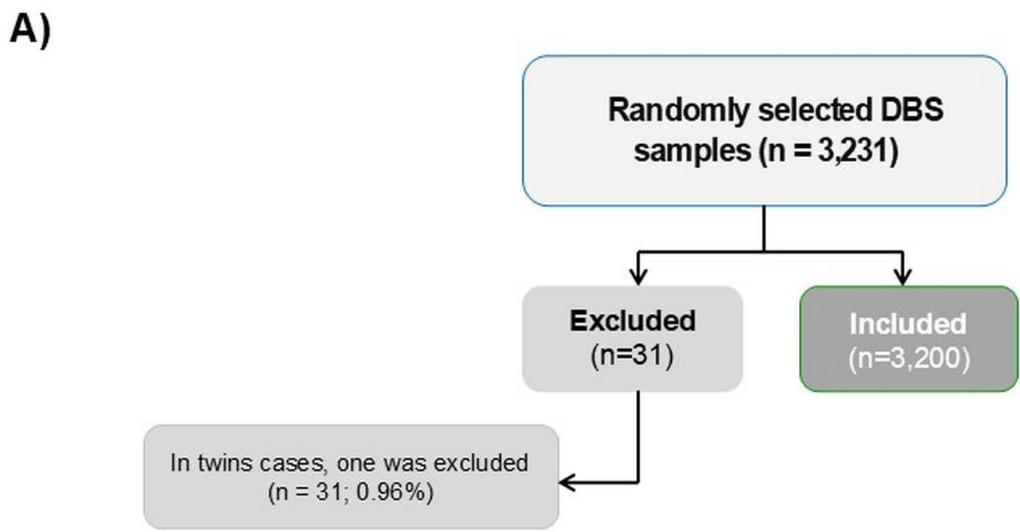

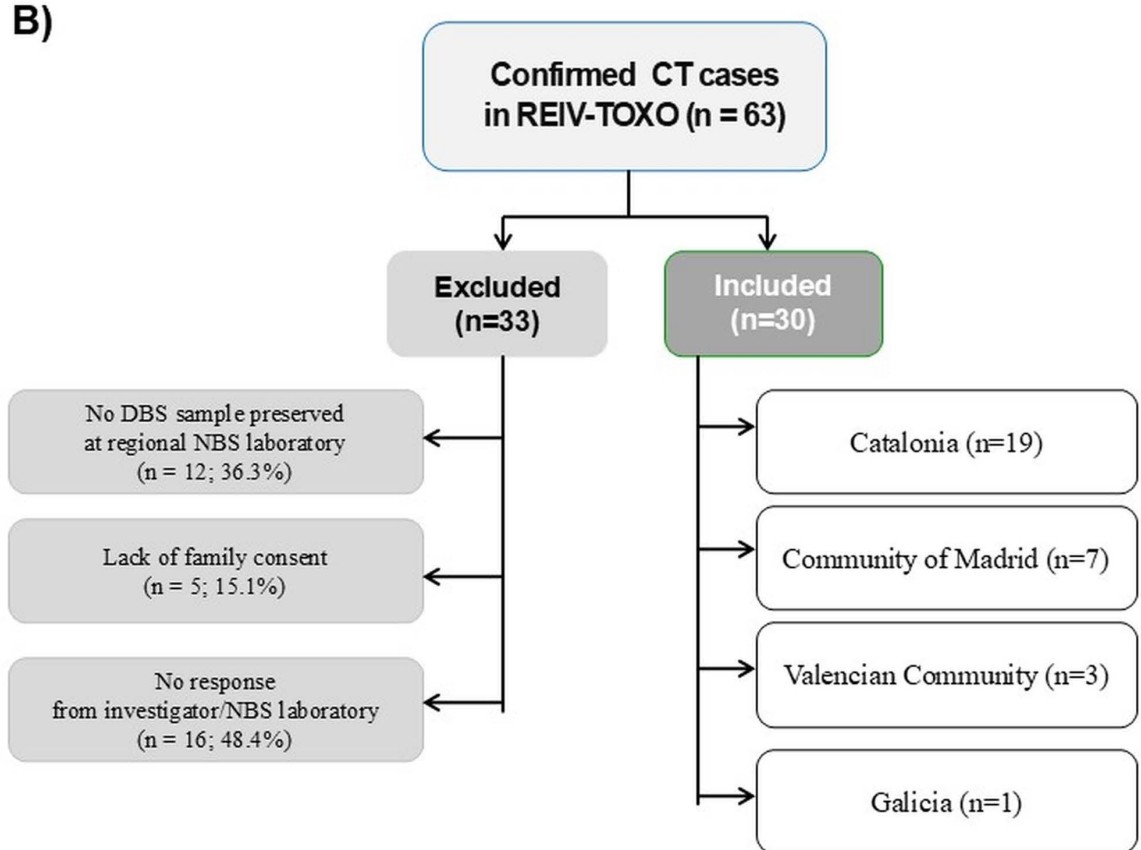

**Fig 2. Flow diagram of selection process of samples.** A) Seroprevalence *T. gondii*-specific Ig G study; B) CT Ig M study. *CT = Congenital Toxoplasmosis; DBS = dry blood spot; NBS = newborn screening program.*

**Table 1. Demographic data and *T. gondii* seroprevalence in the study population.**

| Sample size | n = 3200 % (n) | IgG (+) *T. gondii* % (n) | p-value[1] |
|---|---|---|---|
| All newborns | | 15.5 (499/3200) | – |
| Age of PW | | | |
| < 25 years | 8.5 (271/3,166) | 15.1 (41/271) | |
| 25–35 years | 50.5 (1,598/3,166) | 13.4 (215/1,598) | |
| > 35 years | 41.0 (1,297/3,166) | 18.4 (239/1,297) | **< 0.01** |
| Continent of birth of PW | | | |
| Africa | 10.4 (331/3,169) | 18.7 (62/331) | |
| North America | 0.1 (3/3,169) | 0 (0) | |
| South America | 13.9 (441/3,169) | 30.3 (134/441) | |
| Asia | 4.0 (128/3,169) | 13.2 (17/128) | |
| Europe | 71.6 (2,266/3,169) | 12.4 (282/2,266) | **< 0.01** |
| Spanish-born PW | 65.2 (2,067/3,169) | 11.8 (245/2,067) | |
| Foreign-born PW | 34.8 (1,102/3,169) | 22.6 (250/1,102) | **< 0.01** |
| Area of residence | | | |
| Urban area | 91.0 (2,905/3,191) | 15.5 (453/2,905) | |
| Rural area | 9.0 (286/3,191) | 15.7 (45(286) | 0.95 |

*PW = pregnant women.*

Totals may vary across variables due to missing data in the NBS database.

[1] Chi-square test or the Wilcoxon test were used for group comparisons (p-value calculated at a 95% significance level).

non-consent of the families (5/33; 15.1%), and no response from the investigator or the regional NBS laboratory (16/33; 48.4%).

Only 1/30 (3.3%) DBS samples analyzed tested positive for retrospective *T. gondii*-specific IgM antibodies. In contrast, newborn IgM serology performed at birth revealed a significantly higher detection rate, with 10/27 samples (37%) testing positive (IgM was not evaluated in three samples). Consequently, neonatal screening for *T. gondii* based on retrospective DBS IgM detection identified only 1/10 (10%) confirmed CT cases with positive serum IgM at birth, corresponding to an estimated sensitivity of 10% (95% CI: 0–28.6) and a specificity of 100%. Meanwhile, IgG *T. gondii*-specific IgG antibodies were detected in all 30 cases. The main characteristics of the newborns with processed DBS samples are presented in Table 2.

## Discussion

This study represents the first use of DBS samples to investigate the prevalence of toxoplasmosis in PW in Spain, and additionally assesses their effectiveness for serological diagnosis in confirmed cases of CT. A systematic review study on the reliability of DBS in antibody measurement concludes that there is a good correlation between IgM and IgG concentrations in DBS and serum/plasma samples, supporting its use for seroepidemiological studies like ours [56]. The substantial number of DBS samples from newborns analyzed provided a current demographic picture of toxoplasmosis in PW population in Catalonia considering the lack of published epidemiological data over the past 14 years.

Our findings confirmed a decline in prevalence of *T. gondii* infection among PW in Catalonia, compared to previous data, such as the results of a study carried out in Barcelona (Catalonia) that showed a seroprevalence of 28.6% [57], and studies conducted in other Spanish regions [18–20,58–65]. In Spain, a systematic review and meta-analysis of seroprevalence over the last three decades showed that the pooled seroprevalence for this targeted group was 24.4%

**Table 2. Prenatal history, clinical findings, and IgM results of newborns with CT.**

| CT Cases | Year of birth | Maternal seroconversion | Maternal Treatment | Clinical manifestations | Results of IgM in serum at birth | Results of IgM in DBS samples | Results of IgG in DBS samples |
|---|---|---|---|---|---|---|---|
| 1 | 2015 | 3T | SPI | No | N | N | P |
| 2 | 2015 | 3T | NT | Yes | P | N | P |
| 3 | 2016 | 3T | S | No | NP | N | P |
| 4 | 2016 | 3T | SPI | No | P | N | P |
| 5 | 2016 | 1T | NT | Yes | NP | N | P |
| 6 | 2016 | 2T | SPI | No | P | N | P |
| 7 | 2017 | 3T | SPI | No | N | N | P |
| 8 | 2017 | 3T | SPI, PSA | No | N | N | P |
| 9 | 2017 | 3T | NT | No | N | N | P |
| 10 | 2018 | 2T | PSA | No | NP | N | P |
| 11 | 2018 | 3T | SPI | Yes | P | N | P |
| 12 | 2018 | 3T | PSA | No | P | N | P |
| 13 | 2018 | 3T | NT | No | N | N | P |
| 14 | 2018 | 3T | NT | No | N | N | P |
| 15 | 2018 | 3T | SPI | Yes | P | N | P |
| 16 | 2018 | 1T | NT | No | N | N | P |
| 17 | 2018 | 1T | NT | Yes | N | N | P |
| 18 | 2019 | 2T | SPI, PSA | No | N | N | P |
| 19 | 2019 | 2T | SPI | Yes | N | N | P |
| 20 | 2020 | 2T | NT | No | P | N | P |
| 21 | 2020 | 3T | PSA | Yes | N | N | P |
| 22 | 2020 | 2T | PSA | No | N | N | P |
| 23 | 2021 | 3T | SPI, PSA | No | N | N | P |
| 24 | 2021 | 3T | PSA | No | N | N | P |
| 25 | 2021 | 3T | NT | No | P | P | P |
| 26 | 2021 | 3T | NT | Yes | P | N | P |
| 27 | 2022 | 1T | SPI | No | N | N | P |
| 28 | 2022 | 3T | SPI, PSA | No | N | N | P |
| 29 | 2022 | 3T | PSA | No | P | N | P |
| 30 | 2023 | 3T | NT | No | N | N | P |

CT = congenital toxoplasmosis; N = Negative NP = not performed; NT = No treatment; P = Positive; PSA = pyrimethamine, sulfadiazine, and folinic acid; SPI = spiramycin; T = trimester.

(24,737/85,703, 95% CI 21.2–28.0%) [20]. However, during this period local seroprevalence rates have varied markedly, since studies with different locations and years were included, ranging from 12% in Elche [18] to 63.3% in Gran Canaria [65], suggesting significant differences in prevalence across different geographic areas and an overall prevalence in the country, more closely aligned with recent estimates. Moreover, a progressive decrease in seroprevalence was observed over that period, in line with what has been detected in other Mediterranean countries, with rates decreasing from 27.5% to 21.5% in Italy and from 36.7% to 31.3% in France [1,66–69]. This global decrease in toxoplasmosis prevalence is thought to be the result of changes in eating habits and better hygiene practices during pregnancy.

In our study, the lowest prevalence was observed among Spaniards, whereas the highest was detected among PW of South American origin, followed by those of African origin. These findings are consistent with global epidemiological patterns of *T. gondii* distribution, since Africa and South America have a higher prevalence of the parasite and underscore the

influence of migratory flows on the overall seroprevalence observed in Catalonia [1,8,11,12,20,70]. These variations in *T. gondii* seroprevalence are generally associated with the presence of different climate, lifestyles or economic conditions, with major risk factors including the consumption of raw or undercooked meat, unwashed raw vegetables or fruits, contaminated water, and contact with cats [1,4,6,11,71,72]. In Catalonia, seropositivity rates were low and increased with age, reflecting a lifelong but relatively low risk of infection, largely attributable to good hygiene measures and culinary habits such as increased consumption of frozen meat, as observed in previous studies from low-prevalence areas [1,8,73–75]. However, this also reflects a large proportion of women (84.5%) remain susceptible to acquiring a primary infection during a future pregnancy. In contrast, in high-prevalence areas, the parasite circulation is presumed to be higher, and the acquisition of the infection tends to occur at earlier stages of life. However, in these countries if women have not been exposed to the parasite by childbearing age, their chances of experiencing a primary infection during pregnancy are increased [8,76,77].

Although living in rural areas is considered a risk factor for *T. gondii* seropositivity [11,77], no significant differences between urban and rural areas were observed in our study, but it should be noted that 91% of PW came from urban areas potentially leading to a selection bias. Previous reports depicted high levels of seroprevalence in rural or suburban regions of low resource countries, associated with poorer sanitary conditions, more frequent contact with soil and animals, and the consumption of inadequately treated water [11,77,78]. This association has not been observed in some studies conducted in rural areas of Western countries despite an expected greater environmental exposure, likely due to the high standards of care enforced by legal regulations, comprehensive education for PW, easy access to medical care and established programs for early prenatal diagnosis and treatment [79–80], as occurs in the population analyzed in our study.

Understanding the epidemiological data on gestational toxoplasmosis and CT is crucial for implementing effective public health strategies to mitigate the disease. Despite the decline in seroprevalence among PW in Spain, new cases of CT are detected annually due to the maintenance of prenatal serological screening [13,20,25]. Currently the implementation of this measure varies across different regions of Spain, according to a recent survey conducted by REIV-TOXO [54]. In the absence of a prenatal approach, neonatal screening serves as a tertiary prevention strategy aimed at preventing and minimizing the occurrence of sequelae through the systematic serological identification of *T. gondii* infection at birth in cord blood, peripheral blood or DBS samples, which allows for early treatment of infected newborns **(Fig 1)** [42,43,47,81,82].

Since 1986, newborn screening programs have been implemented on whole blood specimens (cord blood and DBS from heel pricks) routinely collected just after birth in Massachusetts, New Hampshire, Denmark (stopped in 2007), and Brazil [13,42,46,48,83]. This strategy could seem to be attractive since the problems and costs of prenatal diagnosis and treatment naturally do not exist, and could avoid maternal anxiety during pregnancy. Globally, traditional screening tests for specific IgM, such as ELISA and ELISA-like assays, could detect between 44 and 81% of infected newborns at birth [84]. Several factors may contribute to the absence of IgM antibodies at birth, including the immunological immaturity of the newborn, the timing of maternal infection -particularly in late pregnancy-, and the effect of prenatal treatment on microbiological results [4,85]. *Toxoplasma* treatment during pregnancy might play a crucial role in achieving negative microbiological outcomes at birth, which can delay the serological confirmation of suspected CT. This underscores the importance of careful serologic follow-up during the first year of life for newborns with normal workup at birth [13,33,85,86]. On the other hand, false positives results have been reported in all screening tests, requiring validation with reference tests conducted by clinical reference laboratories, often at substantial costs [87–92]. This scenario is expected to improve with the introduction of novel point-of-care tests, particularly IgG and IgM immunochromatographic tests (ICTs). These tests have demonstrated high efficacy in minimizing the likelihood of false positives (IgG and/or IgM) while maintaining maximum sensitivity, which allows prompt follow-up and timely treatment. Although ICTs have not yet been evaluated in NBS settings, their incorporation into prenatal screening programs has been proposed, even in countries with a lower incidence of CT [25,92,93].

Unfortunately, the retrospective DBS samples analysis of infected cases from the REIV-TOXO database, after months or years of storage, did not provide sufficient data to assess the sensitivity of *T. gondii*-specific IgM detection as a

prospective NBS method. This is because it cannot be directly compared to the NBS conducted immediately after birth, as it was performed in all published studies [13,42–46,48–50,52,53,82]. Nonetheless, the low incidence of IgM detection in CT cases, even in serum samples collected at birth, as observed in the REIV-TOXO cohort limits its retrospective use as a NBS tool [25].

In addition to the limitations discussed above, the key issue that turns neonatal screening into suboptimal strategy is the impossibility to offer prenatal treatment. Recent studies have concluded that prenatal treatment, as a secondary prevention measure (Fig 1), effectively reduces MTCT and minimizes clinical sequelae in infected offspring [25–35]. Implementing this measure requires robust diagnostic capabilities to identify all susceptible cases. At present, prenatal screening represents the most effective tool for identifying all children potentially at risk of CT [13,25,68,94]. In our opinion, neonatal screening should only be considered in scenarios where prenatal screening cannot be implemented.

The low detection rate of *T. gondii*-specific IgM antibodies in DBS samples from confirmed CT cases provides valuable insights regarding the effectiveness of the IgM serological test for the retrospective diagnosis of CT after birth. Our results are in concordance with the study conducted by Mangaroni et al. [51], which reported a 40% (2/5) positive rate for IgM Western Blotting (WB) in retrospectively analyzed DBS samples with positive IgM in blood at birth. Despite the use of different serological tests in both studies and the demonstrated higher sensitivity of the WB technique, it is important to emphasize that many factors have contributed to these low sensitivities observed. The analytical performance of laboratory methods, the elution protocols applied, and, most notably, the storage conditions of DBS samples may have significantly influenced the final results [56]. The stability of antibodies is highly sensitive to temperature and humidity. Optimal storage conditions for DBS samples have been demonstrated to be at -20°C in plastic bags with silica desiccant [51,56,81,95]. A study of validation of DBS in toxoplasmosis testing showed 100% sensitivity and specificity for anti-*toxoplasma* IgG detection freshly prepared or storage at -20 °C for 3 months but decrease when they were preserved for more than a month at 4°C or more [23]. In contrast, the Catalonian NBS Laboratory stores samples at 5°C for the first three months, followed by storage at <23°C (under controlled room temperature conditions) for an additional five years. These differences could have impacted the final results obtained. Given these observations, it may be advisable to establish standardized protocols for DBS storage and handling, including long-term storage at −80°C to minimize pre-analytical variability and ensure optimal preservation of antibody integrity for future diagnostic and seroprevalence studies.

Although the low IgM sensitivity may partly reflect sample degradation, prolonged storage could theoretically also reduce IgG detectability, potentially leading to an underestimation of seroprevalence. Nevertheless, this scenario is unlikely, given the inherent stability of IgG, its higher concentration, and the fact that samples were stored for no longer than one year. Consequently, all these factors may explain the 10-fold higher detection of IgG compared to IgM, which ultimately affects their detectability [23,56]. It should be remembered that IgG antibodies detected in the newborn come mainly from the mother and are present at elevated levels, while IgM is produced by the neonate and is usually in low concentration at the beginning of the infection or according to the maturation of the immune system. In addition, all IgG results from DBS samples of confirmed CT cases were successfully detected, even in samples stored up to 103 months earlier. Finally, it is important to emphasize that even in the ideal scenario of testing only well-preserved DBS samples, the same factors that lead to false-negative IgM results in newborns remain significant. As a result, retrospective DBS testing cannot detect all cases of CT and negative IgM results cannot rule out CT diagnosis. To enhance the retrospective diagnosis of CT, *T. gondii* DNA detection through PCR assays could be considered; however, the very low and transient parasitemia [4,13] may limit their diagnostic accuracy. This is supported by the sensitivity of PCR in blood at birth in the Spanish REIV-TOXO cohort, which was only 21.7% (10/46) [25]. To our knowledge, studies of *T. gondii* DNA recovery from DBS samples are very scarce. In the recent publication on the REIV-TOXO cohort in Spain, none of the four DBS samples tested positive for *T. gondii* by PCR [25].

Our study had several limitations. First, the retrospective design and sample selection in the population seroprevalence analysis may influence the results. However, the primary demographic characteristics of the sample aligned with

those of the general population, supporting the validity of our findings. Second, serological testing on DBS samples from CT cases could not assess the test's efficacy as a neonatal screening tool due to the retrospective nature and extended storage time of the samples. Prospective studies that analyze DBS samples immediately after birth are needed to address this question. Third, the study did not include additional tests such as WB, PCR, or the analysis of *T. gondii*-specific IgA antibodies, which might have enhanced retrospective diagnosis in DBS samples from CT cases. Limited sample availability and budget constraints, which focused resources on ELISA-based serological testing, restricted the possibility of exploring these options. Fourth, the small sample size and the very low number of positive DBS samples precluded the performance of univariate and multivariate analysis; therefore, only descriptive analyses were conducted. Last, the varying storage conditions of DBS specimens in the regional NBS laboratories could impact the analysis depending on their origin. However, no data on the specific conditions from each laboratory were provided to the investigators.

In conclusion, the analysis of *T. gondii*-specific IgG antibodies in DBS samples provides valuable insight into the current seroprevalence among PW, revealing a decline in *T. gondii* infection in this population in Catalonia compared with previous studies. Given the existing limitations of neonatal screening, prenatal screening is currently considered the most appropriate approach to identify children potentially at risk for CT, initiate prenatal treatment, and ensure appropriate follow-up. Nevertheless, because our study did not include a direct comparison of different screening strategies; these conclusions should be interpreted with caution. Based on the low IgM detection rate and the strong IgG concordance observed, our findings confirm that DBS are a valid tool for seroepidemiologic surveillance but are unsuitable for the diagnostic screening of CT. Finally, from a public health perspective, these findings highlight the importance of maintaining effective surveillance of *T. gondii* infection and critically evaluating the most appropriate screening strategies to ensure early detection and follow-up of CT. Strengthening evidence-based policies in this area may contribute to improving prevention and long-term outcomes at the population level.

## Supporting information

**S1 Table. *Toxoplasma gondii* IgM assay validation.**
(DOCX)

**S2 Table. *Toxoplasma gondii* IgG assay validation.**
(DOCX)

**S3 Table. Demographic data from Catalonian newborns' mothers.**
(DOCX)

**S1 Appendix. Members of the Spanish REIV-TOXO group.**
(DOCX)

## Acknowledgments

We would like to thank the regional NBS laboratories for kindly providing the DBS samples, and Carmen Martínez for her invaluable contribution to sample processing. We are also grateful to all the patients and families for their participation in REIV-TOXO, in particular the Bentos-Manau family for its unconditional support to the project, as well as all the investigators in the network and Ana Vázquez for statistical support

## Author contributions

**Conceptualization:** Borja Guarch-Ibañez, Ana Argudo-Ramírez, Clara Carreras-Abad, Marie Antoinette Frick, Pere Soler-Palacin, Isabel Fuentes.

**Data curation:** Borja Guarch-Ibañez, Ana Argudo-Ramírez, Clara Carreras-Abad, Marie Antoinette Frick, Pere Soler-Palacin, Isabel Fuentes.

**Formal analysis:** Borja Guarch-Ibañez, Ana Argudo-Ramírez, Clara Carreras-Abad, Isabel Fuentes.

**Funding acquisition:** Pere Soler-Palacin, Isabel Fuentes.

**Investigation:** Borja Guarch-Ibañez, Ana Argudo-Ramírez, Marie Antoinette Frick, Fernando Baquero-Artigao, Pere Soler-Palacin, Isabel Fuentes.

**Methodology:** Borja Guarch-Ibañez, Ana Argudo-Ramírez, Clara Carreras-Abad, Marie Antoinette Frick, Pere Soler-Palacin, Isabel Fuentes.

**Project administration:** Borja Guarch-Ibañez, Ana Argudo-Ramírez, Judit García-Villoira, Isabel Fuentes.

**Resources:** Borja Guarch-Ibañez, Ana Argudo-Ramírez, Pere Soler-Palacin, Isabel Fuentes.

**Software:** Borja Guarch-Ibañez, Ana Argudo-Ramírez.

**Supervision:** Borja Guarch-Ibañez, Ana Argudo-Ramírez, Daniel Blázquez-Gamero, Fernando Baquero-Artigao, Judit García-Villoira, Pere Soler-Palacin, Isabel Fuentes.

**Validation:** Borja Guarch-Ibañez, Ana Argudo-Ramírez, Clara Carreras-Abad, Marie Antoinette Frick, Daniel Blázquez-Gamero, Fernando Baquero-Artigao, Pere Soler-Palacin, Isabel Fuentes.

**Visualization:** Borja Guarch-Ibañez, Ana Argudo-Ramírez, Clara Carreras-Abad, Daniel Blázquez-Gamero, Fernando Baquero-Artigao, Pere Soler-Palacin, Isabel Fuentes.

**Writing – original draft:** Borja Guarch-Ibañez, Ana Argudo-Ramírez.

**Writing – review & editing:** Ana Argudo-Ramírez, Clara Carreras-Abad, Marie Antoinette Frick, Daniel Blázquez-Gamero, Fernando Baquero-Artigao, Judit García-Villoira, Pere Soler-Palacin, Isabel Fuentes.

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
