## [Decision Letter · Decision Letter 0]

11 Aug 2025

Study of seroprevalence of toxoplasmosis in pregnancy throughout Newborn Dried Blood Spots in Catalonia (Spain). Assessment of T. gondii-specific IgM and IgG antibodies in Newborn Dried Blood Spots for the diagnosis of Congenital Toxoplasmosis.

Dear Dr. Guarch-Ibañez,

Thank you for submitting your manuscript to PLOS Neglected Tropical Diseases. After careful consideration, we feel that it has merit but does not fully meet PLOS Neglected Tropical Diseases's publication criteria as it currently stands. Therefore, we invite you to submit a revised version of the manuscript that addresses the points raised during the review process.

Please submit your revised manuscript within 60 days Oct 10 2025 11:59PM. If you will need more time than this to complete your revisions, please reply to this message or contact the journal office at plosntds@plos.org. Please include the following items when submitting your revised manuscript:

We look forward to receiving your revised manuscript.

Kind regards,

Jean Joel Bigna

Guest Editor

Laura-Isobel McCall

Section Editor

Shaden Kamhawi

co-Editor-in-Chief

Paul Brindley

co-Editor-in-Chief

**Additional Editor Comments:**

The editorial team and reviewers agree that the manuscript presents valuable data on Toxoplasma gondii seroprevalence in Catalonia and the use of dried blood spots (DBS) for diagnosing congenital toxoplasmosis (CT). They acknowledge the study's strengths, such as addressing a significant knowledge gap and its potential public health relevance. However, the consensus is that the manuscript requires major revisions before it can be considered for publication.

The main concerns include:

1) The primary scientific concern is the reliability of the dried blood spot (DBS) method. The editorial team recommends performing an internal validation by testing a subset of matched serum samples to prove the DBS results are trustworthy. We also pointed out that the sample size for the congenital toxoplasmosis (CT) group (n=30) is too small to draw strong conclusions. To fix this, we suggest adding confidence intervals and performing a more detailed statistical analysis, with the available patient data.

2) Report this study with appropriate reporting guidelines.

3) Add a flowchart detailing the sample selection process and a table with the clinical characteristics of the CT cases, if available.

The authors should provide strong and robust responses to reviewers' comments before further consideration.

**Journal Requirements:**

1) Please provide an Author Summary. This should appear in your manuscript between the Abstract (if applicable) and the Introduction, and should be 150-200 words long. The aim should be to make your findings accessible to a wide audience that includes both scientists and non-scientists. Sample summaries can be found on our website under Submission Guidelines:

2) We noticed that you used the phrase 'data not shown' in the manuscript. We do not allow these references, as the PLOS data access policy requires that all data be either published with the manuscript or made available in a publicly accessible database. Please amend the supplementary material to include the referenced data or remove the references.

3) Tables should not be uploaded as individual files. Please remove these files and include the Tables in your manuscript file as editable, cell-based objects. For more information about how to format tables, see our guidelines:

https://journals.plos.org/plosntds/s/tables 

5) We notice that your supplementary Tables are included in the manuscript file. Please remove them and upload them with the file type 'Supporting Information'. Please ensure that each Supporting Information file has a legend listed in the manuscript after the references list.

6) We note that your Data Availability Statement is currently as follows: "There are ethical restrictions on sharing data set because it contains potentiallyidentifying and sensitive patient information and this restriction is imposed by theResearch Ethics Committee.To request data please contact the Research Ethics Committee of Hospital Universitaride Girona Dr. Josep Trueta (ceic.girona.ics@gencat.cat) and Ethics Committee of Instituto de Salud Carlos III (registro.general@isciii.es).There is no third party organization that has access to the complete data included inthe article.". Please confirm at this time whether or not your submission contains all raw data required to replicate the results of your study. Authors must share the “minimal data set” for their submission. PLOS defines the minimal data set to consist of the data required to replicate all study findings reported in the article, as well as related metadata and methods (https://journals.plos.org/plosone/s/data-availability#loc-minimal-data-set-definition).

- The points extracted from images for analysis..

7) Please ensure that the funders and grant numbers match between the Financial Disclosure field and the Funding Information tab in your submission form. Note that the funders must be provided in the same order in both places as well.

**Reviewers' Comments:**

Reviewer's Responses to Questions

**Key Review Criteria Required for Acceptance?**

**Methods:**

-Are the objectives of the study clearly articulated with a clear testable hypothesis stated?

-Is the study design appropriate to address the stated objectives?

-Is the population clearly described and appropriate for the hypothesis being tested?

-Is the sample size sufficient to ensure adequate power to address the hypothesis being tested?

-Were correct statistical analysis used to support conclusions?

-Are there concerns about ethical or regulatory requirements being met?

Reviewer #1: The manuscript presents valuable data on the use of dried blood spots (DBS) for retrospective diagnosis of congenital toxoplasmosis and contributes to the ongoing discussion on newborn screening strategies. However, it cannot be accepted in its current form due to several mandatory requirements that must be addressed prior to publication.

-The funding statement is poorly formatted and does not meet PLOS NTDs’ strict requirements

It must be rewritten to include: Author initials who received each grant (e.g., BGI, IFC, PSP), Full grant number, Full funder names and …

-include the full name of the ethics committee(s) that approved the study and the approval code (if applicable).

The scientific content and conclusions are sound and potentially impactful, but the manuscript must undergo major revisions to comply with journal formatting and transparency policies, especially regarding funding and data availability. Once these are corrected, the paper will be suitable for acceptance

Reviewer #2: (No Response)

Reviewer #3: Methodology

It would be helpful to include a flowchart summarizing the sample processing workflow—specifically, indicating how many samples were initially received, how many were excluded (and for what reasons), and how many were ultimately used for IgG and IgM antibody testing. This visual aid would enhance clarity and allow readers to better understand the overall study design and sample selection process.

Reviewer #4: Objectives and Hypothesis

Lines 133–140: The objectives are clearly described, estimating maternal T. gondii seroprevalence from newborn DBS and evaluating DBS diagnostic value for CT. However, no explicit hypothesis is stated. For example, you could add:

“We hypothesized that newborn DBS IgG testing can accurately estimate maternal seroprevalence, while IgM sensitivity for detecting CT is low.”

Study Design

Limitation: Because DBS from CT cases were stored for months to years under varying conditions (Lines 402–404), the results do not represent real-time screening sensitivity. This needs to be explicitly acknowledged in Methods so readers understand the design limitation.

Population Description

Lines 270–283: CT case inclusion is described, but maternal infection trimester, prenatal treatment regimen, and timing of sampling are only mentioned in passing (lines 174–179) and not analyzed in relation to IgM positivity. This information is important for interpreting sensitivity and should be presented in a table and statistically analyzed.

Sample Size

Lines 221–223: The sample size calculation for Aim 1 is clear (n≈3,150 for ±1.5% precision).

No equivalent calculation for Aim 2, the CT diagnostic evaluation. With only 30 cases (Line 270), the sensitivity estimates have wide uncertainty. Authors should add confidence intervals for IgM detection rates to quantify statistical uncertainty.

Statistical Analyses

Lines 224–229: Chi-square and Wilcoxon tests are appropriate. Logistic regression is used for seroprevalence results (Lines 265–266).

Perform and present logistic regression for Aim 2 (IgM detection in CT cases) with maternal/infant covariates such as age, origin, trimester of infection, and treatment.

Also, provide exact p-values in addition to CIs when reporting significant findings.

Data Availability

While patient identifiers are protected, consider sharing an anonymized dataset (e.g., maternal age, origin, serostatus) for transparency and compliance with open-data policy.

Reviewer #5: The authors stated two objectives: (1) to evaluate the utility of DBS samples for estimating the current seroprevalence of toxoplasmosis in pregnant women in Catalonia, and (2) to determine the diagnostic value of serological testing (IgM, IgG) of DBS samples collected at birth. Although the sample size is sufficient for objective (1), it is not for objective (2) (n=30). Nevertheless, this may still be acceptable, given that positive cases of CT are inherently rare.

However, there are major concerns regarding the study design.

1) Reliability of DBS testing – While the authors cited a reference reporting a correlation between DBS and serum testing, the reliability of DBS can vary and is influenced by storage conditions, which were not clearly described in this study. If matched serum samples are available, a subset should be tested to establish the correlation within this dataset. Without such internal validation, the authors cannot confidently interpret the observed decrease in prevalence by comparing it with previous studies that used serum testing.

2) Low IgM positivity in the CT subgroup – Although the authors discussed the differences between IgG and IgM, the low IgM positivity observed in the 30 CT cases further raises concerns about DBS reliability. Moreover, because only DBS samples from IgG-positive serum cases were included, the specificity of DBS testing could not be assessed.

3) Limited clinical applicability – Although not explicitly intended by the authors, DBS cannot replace prenatal screening, thereby limiting the clinical significance of the study. While DBS may serve as a retrospective tool to assess the condition of pregnant women, it cannot be used to prevent vertical transmission.

**Results:**

-Does the analysis presented match the analysis plan?

-Are the results clearly and completely presented?

-Are the figures (Tables, Images) of sufficient quality for clarity?

Reviewer #1: -Tables/Appendices: Ensure S1 and S2 Appendices are correctly labeled and cited in the main text.

-Abbreviations: Define all abbreviations at first use (e.g., PW, CT, DBS, NBS, ICT)

Reviewer #2: (No Response)

Reviewer #3: Results

The title of the article is: "Study of seroprevalence of toxoplasmosis in pregnancy throughout Newborn Dried Blood Spots in Catalonia (Spain). Assessment of T. gondii-specific IgM and IgG antibodies in Newborn Dried Blood Spots for the diagnosis of Congenital Toxoplasmosis" However, according to the results presented, this method was not effective for the diagnosis of congenital toxoplasmosis due to the low detection rate of IgM antibodies when compared to newborn IgM serology performed at birth.

In this context, it would be important for the authors to clarify whether they consider this method to be viable for the diagnosis and detection of congenital toxoplasmosis. At present, this remains somewhat ambiguous, particularly given the apparent.

Reviewer #4: 1. Match Between Analysis Plan and Results

Lines 232–266 (Seroprevalence analysis):

Matches the stated methods (Lines 224–229), prevalence estimates are given overall and stratified by age, origin, and rural/urban status.

Logistic regression is presented for seroprevalence, consistent with the plan.

Lines 270–283 (CT diagnostic evaluation):

Matches the description in Methods (Lines 182–195), assessing IgM and IgG detection in DBS from confirmed CT cases.

No regression analysis is performed for IgM positivity despite this being possible and informative given collected maternal and infant variables.

Include logistic regression for Aim 2 (IgM detection) to fully align with the analytical potential described in Methods.

2. Clarity and Completeness of Results

Clarity Strengths:

Table 1 (Lines 246–247) clearly summarizes demographic data and IgG prevalence.

Key prevalence values are given with 95% CIs, fulfilling statistical clarity requirements for Aim 1.

Differences by origin and age are well-reported (Lines 254–262).

Completeness Issues:

Aim 2 results (Lines 270–283) are sparse, with no confidence intervals for IgM sensitivity or specificity.

CT cohort characteristics (e.g., maternal infection trimester, prenatal treatment, infant clinical presentation) are not fully tabulated despite being partly available from REIV-TOXO. These are important for interpreting why IgM detection was so low (3.3%).

3. Figures and Tables

Figure 1 (mentioned at Line 81) is referred to but not included in the extracted text; ensure it is high resolution, labeled with abbreviations explained in the legend, and interpretable without reference to the main text.

Any flow diagram of sample selection for both Aim 1 and Aim 2 is missing. A PRISMA-style flow chart for CT case inclusion (starting from 63 cases to 30 included) would improve clarity (Lines 270–276).

Recommendation:

1. Verify Figure 1 meets journal quality standards (minimum 300 dpi, vector format preferred).

2. Add a flow diagram summarizing sample selection and reasons for exclusion for both seroprevalence and CT case analysis.

Reviewer #5: The analysis follows the pre-specified plan; however, there are concerns regarding the study design, so the results are not compelling.

In addition, the necessity of Figure 1 for this article is questionable.

**Conclusions:**

-Are the conclusions supported by the data presented?

-Are the limitations of analysis clearly described?

-Do the authors discuss how these data can be helpful to advance our understanding of the topic under study?

-Is public health relevance addressed?

Reviewer #1: (No Response)

Reviewer #2: (No Response)

Reviewer #3: Conclusion

The conclusion effectively addresses the results and their limitations, and it appropriately suggests that improved sample handling could enhance detection methods, particularly given the limited stability of antibodies such as IgM. To strengthen this point, it may be advisable for the authors to propose a standardized protocol for sample storage and handling. This would help ensure that future diagnostic and seroprevalence studies are not hindered by pre-analytical variability affecting antibody detection.

Reviewer #4: The study’s conclusions are largely supported by the data; IgG detection in DBS is useful for estimating maternal seroprevalence, and IgM detection in stored DBS is unsuitable for neonatal screening. However, the claim that prenatal screening is “most effective” should be slightly tempered since the study did not directly compare screening strategies. Limitations are well described but should also note the small CT sample size and statistical imprecision. The work advances understanding by filling a 14-year data gap in Spain and highlighting methodological constraints of DBS use. Public health relevance is addressed but could be strengthened by linking results to screening policy and targeted interventions for high-risk groups.

Reviewer #5: Given the limitations in the study design, the conclusion is difficult to accept.

**Editorial and Data Presentation Modifications?**

Reviewer #1: (No Response)

Reviewer #2: • The title is quite long. I would recommend shortening the title to include the most relevant information (e.g. Using dried blood spots to estimate Toxoplasma gondii seroprevalence in pregnant women and to serologically diagnose congenital toxoplasmosis in Catalonia, Spain)

• Given that there are many serological methods to evaluate T. gondii antibodies, I think it would be helpful to readers to have the type of test used in this study (ELISA) included in the abstract.

• Line 64: add space between “Latin” and “American”

• Line 74: use either “primo-infection” or “primary infection” rather than primoinfection

• Figure 1, Line 91: change “antenatal” to “prenatal” to be consistent throughout the text

• Line 93: remove “)” after “2010”

• Line 94: remove “of” before “12%”

• Line 101: add “region” or “area” after “geographic”

• Line 152: remove “.” before “gender”

• Line 168: you previously defined “immunoglobulin” as “Ig” so you can directly say “IgM or IgA”

• Lines 169, 359, 400: italicize Toxoplasma

• Line 183: again, you previously defined “Ig” so you can simplify to “IgM and IgG”

• Line 195: rather than give a date when the test was performed, perhaps give range of years after sample collection for CT cases (e.g. “and between X and X years after sample collection for positive CT confirmed cases”)

• Lines 221-224: I recommend moving the sample size calculation to section 1.1 so the reader knows right away that the sample size included in the study is appropriate

• Table 1: In the methods you state that 3231 newborns were tested and in the results say that 31 were excluded due to the birth of twins, resulting in a total of 3200 mothers. It is unclear why the denominators in Table 1 vary from this total (3200) and across the different demographic variables examined. Further clarification on how you arrived at the total populations evaluated in each category is needed. In the results you state that only 3152/3200 babies were born to residents of Catalonia. If your goal is to assess the seroprevalence of pregnant mothers in Catalonia specifically, should you not limit your population to residents and exclude mothers that reside outside of Catalonia?

• Line 255: remove “’” from “foreigner’s”

• Lines 260-262: What was the odds ratio for pregnant women <25 years compared to pregnant women >35 years? Or compared to pregnant women 25-35 years? It might be interesting to look at trend across all three age categories.

• Lines 277-278: What were the results for the 5 DSB samples from the CDC IgM program that you used as controls? Assuming that the CDC confirmed the presence of IgM in these 5 cases, it would be interesting to see if they also performed as poorly with this ELISA kit for detecting IgM.

• Lines 282-283: Was IgG evaluated in serum for these CT cases in addition to IgM? You mention that IgG values were tested in the methods (line 176).

• 388-389: I think rather than referring to the “ELISA technique” (since many Toxoplasma ELISA assays exist and only one was tested in this study) you could more generally say “effectiveness of IgM serological testing”, especially given your comparison in the next few sentences to the western blot study.

Reviewer #3: (No Response)

Reviewer #4: (No Response)

Reviewer #5: (No Response)

**Summary and General Comments:**

Reviewer #1: The manuscript presents valuable data on the use of dried blood spots (DBS) for retrospective diagnosis of congenital toxoplasmosis and contributes to the ongoing discussion on newborn screening strategies. However, it cannot be accepted in its current form due to several mandatory requirements that must be addressed prior to publication.

-The funding statement is poorly formatted and does not meet PLOS NTDs’ strict requirements

It must be rewritten to include: Author initials who received each grant (e.g., BGI, IFC, PSP), Full grant number, Full funder names and …

-References: Inconsistencies in journal abbreviations and punctuation (e.g., "J Clin Microbiol" vs "J. Clin. Microbiol." in refs 80 and 94). All references must follow journal style.

-Tables/Appendices: Ensure S1 and S2 Appendices are correctly labeled and cited in the main text.

-Abbreviations: Define all abbreviations at first use (e.g., PW, CT, DBS, NBS, ICT)

-include the full name of the ethics committee(s) that approved the study and the approval code (if applicable).

The scientific content and conclusions are sound and potentially impactful, but the manuscript must undergo major revisions to comply with journal formatting and transparency policies, especially regarding funding and data availability.

Reviewer #2: The article is generally clear and well organized. The authors seek to fill an important knowledge gap regarding regional seroprevalence of the zoonotic parasite Toxoplasma gondii in Catalonia, Spain and the demographic influences on seroprevalence. They also aim to explore the utility of dried blood spots for the retrospective serological diagnosis of congenital toxoplasmosis. The article acknowledges the limitations of retrospective studies and emphasizes the clinical importance of identifying cases of congenital toxoplasmosis and initiating early treatment.

Reviewer #3: In the introduction, it may be helpful to include a brief paragraph explaining the differences between IgM and IgG antibodies. This would provide readers with a clearer understanding of why IgG detection remains a useful tool for estimating seroprevalence, and how the detection of both immunoglobulins can contribute to the diagnosis in newborns.

Reviewer #4: (No Response)

Reviewer #5: This study has the strength of assessing prevalence in a resource-limited region without systemic prenatal screening, utilizing available DBS samples. However, supporting evidence is needed to demonstrate that DBS testing reliably reflects the serological status of pregnant women. In addition, there are concerns regarding selection bias in both the DBS and CT samples.

PLOS authors have the option to publish the peer review history of their article (what does this mean? ). If published, this will include your full peer review and any attached files.

**Do you want your identity to be public for this peer review?** For information about this choice, including consent withdrawal, please see our Privacy Policy .

Reviewer #1: No

Reviewer #2: No

Reviewer #3: No

Reviewer #4: No

Reviewer #5: No

**Figure resubmission:**

**Reproducibility:**



---

## [Decision Letter · Decision Letter 1]

10 Nov 2025

Response to Reviewers
Revised Manuscript with Track Changes
Manuscript

Shaden Kamhawi

co-Editor-in-Chief

Paul Brindley

co-Editor-in-Chief

**Reviewers' comments:**

**Key Review Criteria Required for Acceptance?**

**Methods:**

-Are the objectives of the study clearly articulated with a clear testable hypothesis stated?

-Is the study design appropriate to address the stated objectives?

-Is the population clearly described and appropriate for the hypothesis being tested?

-Is the sample size sufficient to ensure adequate power to address the hypothesis being tested?

-Were correct statistical analysis used to support conclusions?

-Are there concerns about ethical or regulatory requirements being met?

Reviewer #1: there is no explicit hypothesis statement. The objectives imply hypotheses (“DBS can estimate maternal seroprevalence accurately” and “DBS can identify CT cases”), but these should be formally stated in the final paragraph of the Introduction.

The study design aligns well with the objectives, but clarifying the prospective vs. retrospective components in the first paragraph of Methods would improve reader understanding.

more clarity could be added on how “foreign-born” mothers were categorized (by birth country or nationality).

specify how borderline results were treated statistically (they were reclassified as negative, but this should be justified.

There are no ethical concerns; regulatory compliance appears thorough and transparent

Reviewer #3: The revised work addresses the reviewers' comments.

They now clarify their hypothesis and specify their objectives.

The population analyzed is small, but they clarify this point in the limitations section and adequately address the reviewers' comments.

The authors address the study's limitation of not applying another statistical analysis, so this point is clear.

They have a section on ethical considerations.

Reviewer #4: 1. Objectives and Hypothesis

Strengths

The objectives are clearly articulated in both the abstract and introduction.

Weakness

The manuscript lacks a formal testable hypothesis statement. It reads as an observational aim rather than a hypothesis-driven study.

Explicitly state a hypothesis such as : “We hypothesized that DBS-based IgG detection accurately reflects maternal seroprevalence and that DBS-based IgM detection is insufficient for diagnosing congenital toxoplasmosis.”

2. Study Design Appropriateness

Strengths

The cross-sectional design using existing DBS samples from a population-based newborn screening program is appropriate for estimating maternal seroprevalence.

The case-series design using confirmed CT cases from the REIV-TOXO registry appropriately evaluates diagnostic performance.

Ethical approvals and informed consents are reported in detail.

Weakness

The retrospective design for CT case analysis limits conclusions about diagnostic sensitivity, as long storage time may have affected IgM stability.

The design does not allow a direct comparison with serum collected at birth weakening conclusions about neonatal screening feasibility.

3. Study Population

Strengths

The population is well defined: 3,231 DBS samples randomly selected

Demographic variables (maternal age, origin, and area of residence) are comprehensively described.

Inclusion/exclusion criteria are clear (singletons, good sample quality, etc.).

Weakness

The REIV-TOXO CT group (n=30) is small and derived from multiple regions with variable sample handling; this introduces heterogeneity.

The authors did not indicate whether the representativeness of the 5.7% sample of total births was validated statistically.

4. Sample Size and Statistical Power

Strengths

Sample size was calculated a priori, assuming a 24.4% prevalence, targeting ±1.5% precision a strong methodological element.

The final sample (n=3,200) exceeded the minimum estimate (3,150).

Weakness

For the CT diagnostic analysis, n=30 is underpowered to reliably estimate sensitivity/specificity (the confidence interval is wide: 0–28.6%).

Authors acknowledge this limitation, but it still constrains generalizability.

5. Statistical Analysis

Strengths

Appropriate descriptive and inferential statistics were used

Weakness

No adjustment for potential multiple testing or missing data imputation was described.

The authors did not include goodness-of-fit or model diagnostics for logistic regression.

Sensitivity/specificity estimation for IgM was limited by small sample and retrospective bias.

6. Ethical and Regulatory Compliance

Strengths

Ethical approval from multiple committees (Hospital Trueta, Carlos III Institute, etc.) is well documented.

The study followed the Declaration of Helsinki.

Parental consent for use of DBS in CT cases was obtained.

Data confidentiality and controlled storage were observed.

Reviewer #5: Authors should provide point-by-point responses.

1) Reliability of DBS testing

Although authors provided assay validation data, a more detailed explanation is needed. Where did the negative quality control come from? The sensitivity for IgG appears to be 100%, but how about the specificity? This figure is critical for accurately estimating prevalence based on DBS testing.

3) Limited clinical applicability

This comment was not properly addressed in the response letter, although it was reflected in the introduction, which resolved most of the initial concerns. However, another issue remains. The authors concede that the low IgM sensitivity might be due to storage. This logically raises concern that IgG sensitivity could also be compromised by the same storage issues. The 100% IgG sensitivity figure, derived from a small sample of only 30 CT cases, may be an overestimation and not reflective of the test's true performance. If the true IgG sensitivity is indeed lower than 100%, the test would have missed true positives, leading to an underestimation of the reported 15.5% seroprevalence. Although storage issue was stated in the discussion, the chance of underestimation should also be stated.

**Results:**

-Does the analysis presented match the analysis plan?

-Are the results clearly and completely presented?

-Are the figures (Tables, Images) of sufficient quality for clarity?

Reviewer #1: Some numerical details are repeated both in text and tables (e.g., specific p-values and percentages). Summarize trends in the text and leave exact values in tables.

Reviewer #3: The results meet the objectives of the proposed hypothesis.

By responding to the editors' comments, the authors improve the clarity of the presentation of the results.

Reviewer #4: 1. Concordance Between Analysis Plan and Presented Results

Strengths

The analyses presented align well with the objectives and method outlined in the “Materials and Methods” section.

Weaknesses

The authors mention “multivariate logistic regression” but do not provide the regression coefficients, odds ratios, or model diagnostics in tabular form; only summary interpretations are given in text.

There is no explicit comparison of planned versus executed analyses for the CT group; for example, PCR or IgA tests (acknowledged as not performed) deviate from what a complete diagnostic evaluation might entail.

The analysis plan did not specify how missing data were handled (denominator variations are noted but not statistically explained).

2. Clarity and Completeness of Results Presentation

Strengths

Results are presented in logical order

Weaknesses

Some statistical details (e.g., exact p-values, CI bounds for odds ratios, and full logistic model parameters) are absent.

There is no table showing multivariate regression output, only narrative interpretation.

3. Quality and Clarity of Figures and Tables

Strengths

Figures and tables appear in consistent PLOS formatting style with proper legends and abbreviations.

Reviewer #5: (No Response)

**Conclusions:**

-Are the conclusions supported by the data presented?

-Are the limitations of analysis clearly described?

-Do the authors discuss how these data can be helpful to advance our understanding of the topic under study?

-Is public health relevance addressed?

Reviewer #1: Consider slightly reframing the concluding paragraph to emphasize how the evidence directly supports each claim. For instance:

“Based on the low IgM detection rate and strong IgG concordance, our findings confirm that DBS are a valid tool for seroepidemiologic surveillance but unsuitable for diagnostic screening of CT.”

Reviewer #3: The results section supports the proposed hypothesis, and the authors clearly acknowledge the limitations of the study, mentioning them in the conclusion.

The study is relevant to public health and highlights the importance of having an easy and reliable method for testing for toxoplasmosis.

The method has limitations, but these are well explained in the study.

Reviewer #4: The main conclusions are well aligned with the findings

Supported by data but would benefit from nuanced phrasing around external validity.

The limitations section is comprehensive and transparent.

The manuscript meaningfully advances understanding, though broader implications for diagnostic strategy could be more explicitly stated.

Public health implications are clearly emphasized

Reviewer #5: (No Response)

**Editorial and Data Presentation Modifications?**

Reviewer #1: Minor Revision

Reviewer #3: (No Response)

Reviewer #4: (No Response)

Reviewer #5: (No Response)

**Summary and General Comments:**

Reviewer #1: The paper is of high scientific merit, well-structured, and well written. It provides important and novel data that fill a critical knowledge gap regarding the epidemiology of T. gondii infection in Spain—particularly in light of discontinued prenatal screening programs. The combination of population-based and registry-based analyses represents a robust approach that strengthens the study’s public health relevance.

The manuscript is suitable for publication after minor to moderate revision, mainly to improve clarity, strengthen discussion of implications, and refine presentation. No ethical or publication concerns are identified.

Reviewer #3: (No Response)

Reviewer #4: (No Response)

Reviewer #5: (No Response)

PLOS authors have the option to publish the peer review history of their article (what does this mean? ). If published, this will include your full peer review and any attached files.

**Do you want your identity to be public for this peer review?** For information about this choice, including consent withdrawal, please see our Privacy Policy .

Reviewer #1: No

Reviewer #3: No

Reviewer #4: No

Reviewer #5: No

**Figure resubmission:**

**Reproducibility:** To enhance the reproducibility of your results, we recommend that authors of applicable studies deposit laboratory protocols in protocols.io, where a protocol can be assigned its own identifier (DOI) such that it can be cited independently in the future. Additionally, PLOS ONE offers an option to publish peer-reviewed clinical study protocols. Read more information on sharing protocols at https://plos.org/protocols?utm_medium=editorial-email&utm_source=authorletters&utm_campaign=protocols

---

## [Decision Letter · Decision Letter 2]

19 Dec 2025

Dear Mr. Guarch-Ibañez,

We are pleased to inform you that your manuscript 'Using dried blood spots to estimate Toxoplasma gondii seroprevalence in pregnant women in Catalonia, Spain, and to serologically diagnose congenital toxoplasmosis”. 

 ' has been provisionally accepted for publication in PLOS Neglected Tropical Diseases.

Best regards,

Laura-Isobel McCall

Section Editor

Laura-Isobel McCall

Section Editor

Shaden Kamhawi

co-Editor-in-Chief

Paul Brindley

co-Editor-in-Chief

We provide below the responses from the last round of reviews for your information.

Reviewer's Responses to Questions

**Key Review Criteria Required for Acceptance?**

**Methods**

-Are the objectives of the study clearly articulated with a clear testable hypothesis stated?

-Is the study design appropriate to address the stated objectives?

-Is the population clearly described and appropriate for the hypothesis being tested?

-Is the sample size sufficient to ensure adequate power to address the hypothesis being tested?

-Were correct statistical analysis used to support conclusions?

-Are there concerns about ethical or regulatory requirements being met?

Reviewer #1: the objectives of the study clearly articulated with a clear testable hypothesis stated.

the study design appropriate to address the stated objectives.

the population clearly described and appropriate for the hypothesis being tested.

the sample size sufficient to ensure adequate power to address the hypothesis being tested.

yes statistical analysis used to support conclusion.

there are not concerns about ethical or regulatory requirements being met.

Reviewer #4: (No Response)

**Results**

-Does the analysis presented match the analysis plan?

-Are the results clearly and completely presented?

-Are the figures (Tables, Images) of sufficient quality for clarity?

Reviewer #1: yes

Reviewer #4: (No Response)

**Conclusions**

-Are the conclusions supported by the data presented?

-Are the limitations of analysis clearly described?

-Do the authors discuss how these data can be helpful to advance our understanding of the topic under study?

-Is public health relevance addressed?

Reviewer #1: yes

Reviewer #4: (No Response)

**Editorial and Data Presentation Modifications?**

Reviewer #1: Manuscript PNTD-D-25-00827R2

Title: Using dried blood spots to estimate Toxoplasma gondii seroprevalence in pregnant women in Catalonia, Spain, and to serologically diagnose congenital toxoplasmosis

Overall Assessment

This is a well-conducted, timely, and clearly written study that addresses an important public health issue

Comments:

The central claim—that DBS IgM is unsuitable for CT diagnosis—is supported by the very low detection rate (1/30). However, storage conditions varied widely (1–103 months, ambient or suboptimal temperatures), which likely degraded IgM.

The authors appropriately note this limitation, but the manuscript should more strongly clarify that these results do not reflect the performance of prospective DBS screening conducted under optimal conditions (e.g., within days of birth and stored at −20°C). The conclusion about neonatal screening feasibility should be more precisely worded to reflect this distinction.

-While acknowledged as a limitation, the omission of IgA (which often outperforms IgM in CT diagnosis) and T. gondii PCR reduces the comprehensiveness of the diagnostic evaluation.

-Given that PCR on DBS is technically challenging but potentially valuable, even negative PCR results from the 30 CT cases would strengthen the argument against DBS-based molecular screening.

-If feasible, consider adding PCR data on a subset of DBS—even if negative—to reinforce the point about low parasitemia. If not, the current discussion is acceptable.

--Add a supplementary table (or a concise main table) showing odds ratios, 95% CIs, and p-values for key variables (maternal age, origin, residence) in the final model.

Minor Comments

Line 161: Typo: “Acurately determination” → “Accurate determination”.

Line 245–247: The decision to reclassify borderline results as negative is justified but may slightly underestimate seroprevalence. Consider briefly discussing the potential magnitude of this bias (e.g., how many borderline results were observed?).

Table 2: Clarify column headers (e.g., “Maternal seroconversion” should specify trimester as “1T = 1st trimester”, etc., which is done in the footnote but could be clearer in the table title).

Author Summary (Line 75): “South America and Africa” — but Table 1 shows South America (30.3%) > Africa (18.7%). Consider saying “especially from South America” for precision.

Reviewer #4: (No Response)

**Summary and General Comments**

Reviewer #1: The central claim—that DBS IgM is unsuitable for CT diagnosis—is supported by the very low detection rate (1/30). However, storage conditions varied widely (1–103 months, ambient or suboptimal temperatures), which likely degraded IgM.

The authors appropriately note this limitation, but the manuscript should more strongly clarify that these results do not reflect the performance of prospective DBS screening conducted under optimal conditions (e.g., within days of birth and stored at −20°C). The conclusion about neonatal screening feasibility should be more precisely worded to reflect this distinction.

Reviewer #4: (No Response)

PLOS authors have the option to publish the peer review history of their article (what does this mean? ). If published, this will include your full peer review and any attached files.

**Do you want your identity to be public for this peer review?** For information about this choice, including consent withdrawal, please see our Privacy Policy .

Reviewer #1: No

Reviewer #4: No

---

## [Editor Report · Acceptance letter]

Dear Mr. Guarch-Ibañez,

We are delighted to inform you that your manuscript, "Using dried blood spots to estimate Toxoplasma gondii seroprevalence in pregnant women in Catalonia, Spain, and to serologically diagnose congenital toxoplasmosis”. 

 ," has been formally accepted for publication in PLOS Neglected Tropical Diseases.

Best regards,

Shaden Kamhawi

co-Editor-in-Chief

Paul Brindley

co-Editor-in-Chief
